# Gender-Related Violence: What Can a Concept Do?

**Pam Alldred**

School of Social Sciences, Nottingham Trent University, Nottingham NG1 4FQ, UK; pam.alldred@ntu.ac.uk

**Abstract:** This paper explains the logic for using the concept of gender-related violence (GRV) as a broad category that problematises homophobia, transphobia and the policing of gender norms and the gender binary, as well as gender-based violence—understood as primarily violence against women and girls (VAWG). It then evaluates the utility of this concept and its capacity to introduce theoretical refinement to the study of gendered violence, by reviewing its reception within a large international feminist project on gendered violence in the lives of children and young people. The aim of this study was to improve knowledge and understanding of forms of violence and discrimination among practitioners who have everyday contact with general populations of children and young people. It sought to improve their ability to identify and challenge sexist, sexualising, homophobic or transphobic language or behaviour, and their knowledge of how and when to refer children and young people to appropriate support. The paper reports my view of the contribution that the concept of gender-related violence made in each of the four project sites: Ireland, Italy, Spain and the UK. The findings are mixed: in Spain and the UK the concept seemed helpful, but in Italy and Ireland it was initially expected to be helpful but, in practice, a conceptualisation closer to *gender-based violence plus homophobia* was employed. It is tentatively concluded that where LGBTQIA+ rights were well-established as well as the problematisation of VAWG, this framework was successful, but that it was less successful in more heteronormative sites where homophobia was less problematised. It is suggested that, as a concept, GRV can make a valuable intervention in sites like the former.

**Keywords:** gender-based violence; gender binary; gender inequality; homophobia; gender-related violence; transphobia

## 1. Introduction

The need for feminist activism to defend women's and girls' rights to live without violence or harassment and to support survivors persists into the present, as commentators including UN Woman and UN Health Promotion have noted. In tandem, harassment and violence toward LGBTQIA+ people remains despite the rights and legal status acquired, for instance, through the European Convention on Human Rights and legislation in some national jurisdictions. At the time the European project described here was being developed, the need to address gendered violence in children and young people's lives was becoming more widely recognised in the UK (Barter 2009; Barter and Berridge 2011; Berelowitz et al. 2013; Fox et al. 2013; Pearce 2009).

This paper describes the broad concept of 'gender-related violence' (GRV) and sets out the context within which it emerged within the author's research, which included large international projects addressing aspects of gendered violence. It suggests that this conceptual framing offers certain opportunities. In a nutshell, GRV as a concept is located within third-wave feminism and queer theory, offering a critical perspective upon not only violence toward women and girls (VAWG), but also heteronormativity, homophobia and the contemporary gender order. As such, it can supply a theoretical framing that articulates with feminist and LGBTQIA+ activism, gender and sexualities research and intersectionality studies.

To assess the claim that conceptually GRV offers these opportunities, the paper goes on to evaluate how the concept was applied in an EU co-funded (DAPHNE III) Action Project

that extended over multiple European countries: the GAP WORK Project. Its full title was Improving Gender-Related Violence Intervention and Referral Through Youth Practitioner Training (Alldred et al. 2014). This project aimed to empower professionals working in everyday roles with children and young people (rather than in specialist violence-support services) to tackle gender-related violence in young people's lives[1].

The paper begins by outlining the general context within which contemporary violence related to gender and sexualities occurs among children and young people. Following this contextualisation, the concept of GRV is introduced. The remainder of the paper assesses the reception of the concept in four of the sites (countries). It concludes by reviewing the differing ways in which the concept operated, and offers initial thoughts on how divergences in application relate to country-specific policy and practice.

## 2. Background: Violence in Children and Young People's Lives

Gendered violence (or what will be called gender-related violence in what follows) and the impact it has on children and young people has attracted increased political and academic attention in recent years in the UK, with various waves of concern focused on peer-on-peer abuse, abuse in interpersonal relationships (IPV), gang-involvement, sexual harassment in schools and on the streets, sexual exploitation of children (CSE) and knife crime (Barter 2009; Barter and Berridge 2011; Berelowitz et al. 2013; Ellis 2004; Firmin et al. 2016; Firmin and Pearce 2016; Pearce 2009; Plan-UK 2016; Reid 2017; Sundaram 2014; Women and Equalities Committee 2018; Wood et al. 2011). Authoritative reports showed that one in three girls experience unwanted sexual touching at school in the UK (Women and Equalities Committee 2018), and 33% of 13–17-year-olds in intimate relationships experienced sexual violence from a partner (Davis 2012). Furthermore, gendered violence began to be highlighted within gang culture, with reports of young women raped in gang-related conflict, and studies of young people's attitudes showed (i) tolerance of interpersonal violence if perpetrated by men within an intimate heterosexual relationship (McCarry 2010); (ii) that young people (e.g., 13–14-year-olds) were less likely to recognise non-consensual sex than older age groups (Coy et al. 2013); and (iii) that boys (still) objectified girls strikingly (Thiara and Coy 2014). International studies provided worrying evidence regarding young people, of violence in their peer relationships, normalised double standards for sexual behaviour (Barter et al. 2015) and poor recognition of everyday sexism (Biglia and Velasco 2012; Lombard 2016). These various studies confirmed that domestic abuse (violence or abuse within intimate relationships) could not be considered only an adult problem (Barter 2011; Barter and Berridge 2011; Ellis 2006), and that multiple forms of violence and abuse affect young people, on- and offline.

Within this wave was an increased awareness of violence among young men (e.g., Reid 2017; Seal and Harris 2016; Whelan 2013) and of the sexual exploitation of boys and young men, but recognition that they were invisible in official discourses and sometimes overlooked in practice (McNaughton Nicholls et al. 2014). At the time, the UK's Rotherham Inquiry (Jay 2014), and soon after another Serious Case Review, the Newcastle Inquiry (Spicer 2018), showed professionals failing to intervene in sexual exploitation. We wanted to ensure that professionals with day-to-day contact with young people recognised and would respond to gendered violence, or sexual violence affecting anyone, and had thought about the norms and stereotypes that could impede their responses. These reviews showed how racism, including in professional responses, might intersect with age and class vulnerabilities, or gendered assumptions about victimhood in the case of boys—each reducing or delaying problematisation and professional intervention. A recent systematic review shows the persistence and breadth of these occlusions (Brady et al. 2022).

For LGBTQIA+ young people, violence, abuse and harassment in UK schools had been both evidenced and acknowledged in policy (Allan et al. 2008; DfES 2002; Rivers 2000; Stonewall 2017) but radical critique was limited by the bullying discourse which centres on the individuals concerned, rather than the cultural norms and values (Alldred and David 2007; Monk 2011). Once norm-driven violence was acknowledged, it became

possible to see the violence of social norms and that the 'everyday culture' was important to problematise in schools (Atkinson 2021; Epstein et al. 2003; Sundaram 2014) and youth settings (Formby 2015; Seal 2016). So, the pejorative use of 'gay' that Stonewall reported 86% of young people hearing in UK schools comes to be problematised as part of an oppressive norm. Furthermore, the critique of cultural norms enabled institutional violence to be recognised. After racist norms and institutional racism had finally been recognised in the UK (MacPherson 1999), institutional heteronormativity could be recognised and seen as a cause and an effect of violence against LGBTQIA+ people, across youth and community settings, schools and universities (Atkinson 2009; DePalma and Atkinson 2009; Epstein et al. 2003; Formby 2015; Seal 2019; Jackson and Sundaram 2020). Thus, the GAP Work Project theoretical framing was to problematise cultural and organisational norms, rather than seeking to highlight individual bullying experiences.

## 3. Development of the Concept of Gender-Related Violence (GRV)

Feminist activism around domestic violence has needed to focus on men's abuse of women and so has tended to work with a concept of violence against women and girls (VAWG) or of gender-based violence (GBV). GBV has been conceptualised as any form of violence that results in physical, sexual or psychological harm to women (World Health Organisation 2002) and can be viewed as a 'slow burn' disaster (Bartoli et al. 2022) or as the "most pervasive human rights violation, transcending class, religion and culture and remaining one of the most serious threats to the health and safety of women and girls worldwide" (Tappis et al. 2016, p. 32; cf Chazovachii et al. 2022).

Gender-based violence (GBV) and misogyny are important concepts and do valuable work in naming long-standing injustices and their impacts on women and girls. GBV highlights the structuring of violence against a class of people, socially and economically positioned and therefore helpfully de-individualised. What the concept of GBV can be understood as doing is a theoretical aggregation of women and girls (Fox and Alldred 2022). Gender-related violence incorporates GBV, but problematises a wider terrain. This broad definition puts gender norms and normativity at the centre. Furthermore, it enables the drawing together of two strands of activism in Western Europe that, historically, were mostly separate: efforts to challenge VAWG and to tackle homophobia, and which have had different legal remedies (Alldred and Biglia 2015).

In 1993, El-Bushra and Piza Lopez (1993) defined gender-related violence as: 'violence which embodies the power imbalances inherent in a patriarchal society' and noted that this is overwhelmingly, though not necessarily, men's violence against women. Over the intervening years, terminology has varied (and see Bufacchi 2005; Frazer and Hutchings 2019), but by 2012, the term *gender-based violence* (GBV) became used more frequently in the anglophone world than 'gender-related violence' (GRV), for instance, in the work of the European Institute for Gender Equality (EIGE). The EIGE definition of GBV is 'violence directed against a person because of that person's gender or violence that affects persons of a particular gender disproportionately.' (What is gender-based violence?)[2].

El-Bushra and Pisa Lopez's definition appealed because of its plurality and socio-cultural analysis. Plurality was valued in order to fully recognise the intersecting power imbalances and the differential impacts that raced, classed etc. hierarchies have on multiply, socio-culturally positioned individuals. A social level of analysis was required in order to problematise the social norms, tolerance and silences around violence and around gender more broadly, rather than define violence narrowly and locate it in the behaviour of problem individuals.

If gender-based violence is founded on gender inequality, then gender-related violence needs to be defined more broadly, to acknowledge other forms of inequality and stigma as well. A tentative definition of GRV was set out in an early project document that is described in more detail in the next section:

*Sexist, sexualising or norm-driven bullying, harassment, discrimination or violence whoever is targeted. It therefore includes gender, sexuality and sex-gender normativities, as well as violence against women and girls* (Alldred 2013).

Gender-related violence (GRV) was understood as:

*Violence that relates to gender, but is not only structured by the primary axis of gender inequality and might include violence (actual, threatened or symbolic) that is enabled by the very concept of gender and so recognises gender normativity, the insistence of a gender binary, homophobia, transphobia, as well as injuries of women's inequality to men—sexism, misogyny, sexual violence and sexual harassment or coercion. (ibid.)*

A motivation for the empirical studies of gendered violence I have undertaken has been to ask: what difference does it make to shift the focus to the 'what' of the problem, not the 'who' is (usually) impacted, and to employ the concept of GRV? The following section describes a project within which this definition was trialled.

## 4. The Project

The impetus for creating an educational package for generic 'youth practitioners' came from my teaching in universities on Youth and Community Work and Initial Teacher Education courses in the UK, and realising where professional education could be strengthened. Hearing accounts from youth workers studying for a Master's degree gave insights into the experiences of some of the young people they worked with. Students asked difficult questions about support and referral for young women, when generic services (say around domestic abuse or sexual violence) might not meet the needs of under 18s, or when age would be a pivot for different professional responses (see e.g., Brady et al. 2022) and young people fear their relationship/s being criminalised. We ended up talking in class about preventative work, educational interventions for with young people, as well as the responsive work youth workers need to be able to do. Consequently, my aim was to produce educational packages covering response to and prevention of gender-related violence, that located the problem culturally, offered value-based and theoretically informed education - not merely training (Cullen and Whelan 2021; Jones et al. 2021) - and to present this, a 'feminist sociology masterclass' [sic] that incorporated 'feminism 101' in the words of a team-member (i.e. started from first principles), in a way that was useful to practitioners.

The intersection of age with gender and sexuality was the particular concern of the project, given its development in the education of youth workers and teachers ('youth practitioners' as we termed the international collective since at international level it encompassed a wide range of roles). The inspiring *No Outsiders* project had challenged heteronormativity in primary schools and called for children to be allowed to question their sexual or gender identity (Allan et al. 2008; Atkinson 2009; DePalma and Atkinson 2009), and research, especially Renold (2005, 2010), had upset romantic assumptions of young children's unquestioning adoption of their given place in the gender order. Feminist scholars were theorising gender in relation to classed and racial inequalities and thinking intersectionally to understand particular forms or sites of violence or inequality (Brah 1996; Grosz 1995; Hill-Collins 2008; Hooks 1984; Lorde 1984; Phipps 2009, 2018; Phoenix and Pattynama 2006; Phoenix 2009; Puar 2007; Skeggs 1997; Sundaram 2014), but few were thinking intersectionally about violence in the lives of children and young people. My teaching showed the need to understand how age might intersect with gender, sexuality, class and race in children's experiences of violence, and for a conceptual framework to enable professionals working with children and young people to be able to identify, problematise and challenge violence regarding gender and sexuality and the inequalities that sustain them. This application to practice propelled the project.

The GAP Work Project (2013–2014) was an EU co-funded (DAPHNE III) Action Project to help 'youth practitioners'—professionals working in everyday roles with children and young people, rather than in specialist, violence-support services—to tackle gender-related violence in young people's lives. It brought together an international group of feminist

activist–academics who were concerned with the lack of training and support available for professionals working with young people (Alldred 2013; Alldred et al. 2014).

This project explicitly built on two influential previous projects: the UK-based, 2006-09 ESRC-funded No Outsiders project mentioned above, and the EU-FRC co-funded AHEAD (Against Homophobia: European Local Administration Devices) project which mapped good practice in tackling homophobia and transphobia in EU countries (see Coll-Planas et al. 2011) in which I and some other GAP Work partners had collaborated. It also drew inspiration from Freirean pedagogy and feminist uses thereof, since I and other UK team members (Cullen, Cooper-Levitan, Whelan) were youth work educators and informed by this (Cooper-Levitan 2023; Cullen 2013), and given my background in sex and relationship education research, the norm critical pedagogies being applied there were also a key element of the project design (e.g., Bromseth and Darj 2010). A mutual educational need was identified between victim-support services (NGOs, third sector and campaign organisations) and youth practitioners in-service or in-training. The aim of our training of practitioners was two-fold. First, to improve practitioners' knowledge of support organisations and legislation and hence their effective referring. Second, to enhance their capacities to challenge violent or discriminatory language and behaviour through making preventative interventions, thereby contributing to the development of a protective environment for children and young people. In return, youth practitioners were a new audience for whom the NGOs could develop training or briefings (e.g., Understand, Identify, Intervene: Supporting young people in relation to peer-on-peer abuse, domestic and sexual violence—Rights of Women) or get feedback on their existing resources. The project therefore brought together partners who were educators/trainers in the third sector, and membership organisations or employers of youth practitioners and social welfare professionals.

Partners in Ireland, Italy, Spain and the UK each developed and piloted their own, contextually relevant training. Each partner shared their training resources, a report of trainees' responses to the training and their evaluation of its success (now hosted on the USVreact/eu website) and published in their own language/s. Associate (unfunded) partners in Hungary and Serbia also fed in their expertise on tackling homophobia and VAWG respectively and had resources translated to pilot there too.

The project sought to bridge gaps between:

- survivors' support services for adults and for children
- victim-support services and everyday professional contact
- supporting those affected and intervening to challenge or pre-empt violence
- interventions tackling dating violence or homophobia.

This fourth gap led to centring the critique of gender normativity in the hope that it would undermine violence against women and girls, as well as homophobic, lesbophobia and transphobic or gender-norm-related violence. The purpose of the project was to design and evaluate training programmes that could bridge these gaps.

The project was framed at the outset (in the funding bid) as a wide and inclusive one since the notion of GRV had grown out of my experience in UK secondary schools where I was delivering workshops for young people. It was the recognition that whilst the workshops to tackle violence against women and girls (VAWG), and the sessions to tackle homophobia were separate initiatives by different campaigning organisations, in theory at least, they shared some common issues: the need to problematise actual and symbolic power in the forms of sexism, heterosexism, machismo, and normativities such as within gender, the gender binary and heteronormativity.

By problematising gender norms, some of the cultural values and assumptions underpinning both these forms of oppression are challenged. The Gay Liberation Movement in the UK, from where the project was conceived, was arguably patchy in its problematisation of gender and support for women's struggles in the 1970s, although there were always some who made the links. About 15 years later, the struggle over the representation and care demanded by HIV/AIDS is seen as a key mobiliser, and in UK cultural politics, resistance to the Criminal Justice Act (1994) helped to undermine a politics based on identity

categories, and importantly, not only in academic seminars. Each country and region has its own story of the relationship between the Gay and Women's Liberation movements, and, no doubt, stories within stories, and of the relevance of feminist political theory to local activism.

## 5. Positioning and Conceptualising GRV

Critique of the devaluing or abjection of the feminine across or within genders and among sexualities creates hierarchies among subjects and relationships (e.g., Vance 1984; Serano 2007). My concern with this came from my cultural studies training and my earlier research on how value judgements about relationships and sexualities create(d) hierarchies regarding who was scrutinised and who was deemed 'fit to parents' (Alldred 1999). Personal experience also informed the critique of monogonormativity and heteronormativity. The femmephobia (Hoskin 2019) seen in these forms of scrutiny or stigma needs to be racialised and can be understood through the concept of misogynoir (see e.g., Davis 2022) and the histories of colonial restrictions on polyamory. Unnamed forms of inequality are created through the monogonorms, and the other oppressive systems that deem who is considered attractive (Harrison 2019) and what forms of relationship face stigma (Klesse 2014; Pallotta-Chiarolli 2010).

Within the context of the project objectives above, it was hoped that conceptually GRV would problematise the gender order underpinning gender violence in a deconstructive move, rather than merely revaluing the devalued side of a binary. The team were committed to including intersectional matrices of domination and inequality, particularly around race, ethnicity and class (Hill-Collins 2017; Phipps 2009; Phoenix and Pattynama 2006). It was also important that this was a more political definition of violence than that used by the World Health Organisation at the time, certainly in the sense that its 2002 definition referred to 'intentional use of physical force or power', and which despite recognising 'community' as a source of violence, was then divided into categories of 'individual' or 'stranger', so that only individual violence was recognised, and not the social, cultural, political or epistemic forms of violence that we wanted to capture.

The first conceptual step had been the recognition that homophobia is often actually a gender slur, frequently an insult to men or boys about their lack of masculinity (perceived, actual or risk of) or of the 'right kind of' masculinity, or to 'unfeminine' women. It could certainly not be understood as affecting lesbian or gay people only, and was harassment of lesbians best understood as homophobia or misogyny? (Lorde 1984). A second was the recognition of biphobia, lesbophobia and femmephobia within the LGBTQIA+ community itself. Again, identity politics failed to protect. The third conceptual requirement was to understand violence in the context of inequalities, and to attend to the intersecting forms of inequality that youth workers described in the lives of the children and young people they worked with: notably age, class (poverty and poverty stigma in particular), gender, sexuality, race and racialisation.

While recognition of the contribution that inequalities make to violence is well-established in feminist thought (after Kelly 1988), education *for equality* and educational interventions *against* gendered violence sometimes diverge, and fail to link abuse and violence relating to gender and sexual orientation to inequalities and cultural norms (Alldred and David 2007). The recognition that gender norms serve to justify and sustain violence led to a broadly framed project that tried to incorporate violence and inequality regarding gender and sexual orientation and to present this in a way that was useful to practitioners. Norm criticality reflected the critical orientation of third-wave feminism and queer theory toward the contemporary gender order and once the project received funding, employing a researcher from Sweden gave us extra confidence in applying it since she read some of its original formulations (e.g., Bromseth and Darj 2010) so it informed the training created, especially by the UK team[3].

The project was devised with Dr Barbara Biglia, whom I had met through critical psychology and feminist activism[4], so shared a commitment to avoiding individualising social

problems. As feminist educators and activists, we wanted an approach that demonstrates the links between different forms of violence, and between violence and power relations, and that foregrounds a political analysis of and response to violence. When we first reached for a term, GRV felt more in line with our theoretical influences in Butlerian queer theory and in more general 'third-wave' (post-structuralist/deconstructive and anti-essentialist) feminisms, our concern to recognise difference and intersectionality within feminist theory, and our activism. We felt that it resisted essentialising because it was not based *on* gender, but rather on any mobilisation of gender (as if it already had 'scare quotes' on 'gender'). Not defining a form of violence in relation to who its victims were enabled keeping an open mind about who would be affected and how, as well as a promise to support anyone so affected. It saw through our rejection of identity politics, problematised the normative behaviour and expectations, rather than rescuing the person wounded by them (Youdell and Armstrong 2011 and insisted on equalities education for social justice, not just for the wellbeing of the (young) people present (Alldred and David 2007).

This culturalist, rather than individualist approach positioned it decisively within sociology rather than psychology, and so resisted the psychological framing of social issues such as the attribution of violence or sexual violence to particular men as perpetrators who might be profiled and identified, and here specifically psychological framing might otherwise draw us back onto the terrain of 'bullying' which despite important interventions made under its name, we had sought distance from because of its binary (bully/victim) framing, and individualising tendencies (in analysis and response).

A broad concept of GRV is compatible with feminist approaches that problematise all inequality and attend to power differentials across social differences (race, ethnicity, class, gender and sexual orientation among them) and emphasise their intersections and accompanying occlusions or silencing (Anthias and Yuval-Davis 1993; Brah 1996; Strid et al. 2013; Walby et al. 2012). Problematising the gender binary and gender and sexual normativities reflects the deconstructive move of 'third-wave' feminism and queer theory in particular. Butler's (1990) articulation of the heterosexual matrix and gender binary's mutual constitution is a key influence in the project's challenge to heteronormativity. In terms of today's cultural politics, it is like preferring to use 'content warning' over 'trigger warning' because it points to the cause (the content) not a presumed effect (a triggering), in order to presume as little as possible and keep the category wide.

When it comes to children and young people's experiences of gendered and sexualised violence, the concept of GRV offers a critical perspective on a range of other framings of gendered and sexualised violence. Bullying was perhaps the dominant framework being used by other researchers, although in the activism and school-based interventions I was part of, a radical equalities framing meant that feminism and LGBTQIA+ equalities were at the base. My experiences of delivering training on equalities issues has confirmed multiple times that positive approaches, as opposed to raising sympathy for 'victims', are more effective and less likely to evoke a defensive reaction in some participants.

Other researchers, including those researching schools, focused more on 'bullying' and Ringrose (2008, p. 518) highlighted its 'considerable political purchase' in the UK, which remains true. Some work on sexual bullying recognised the significance of hegemonic masculinity and of socially constructed gender and sexual binaries (e.g., Meyer 2008) and Ian Rivers whose work has highlighted how gender and sexuality norms create the conditions for bullying (Rivers 2000) and can predict victimisation (Felix et al. 2009).

Turner-Moore et al. (2022) writing about a similar, contemporary EU project argue for uniting sexualised, sexuality and gender expression bullying or harassment under the term sexual bullying. In practice, this term covers the same range of issues as GRV, but retains the reference to bullying. More recently Turner-Moore et al. (2022, p. 90) reported that:

> 'Some researchers have drawn together (i) sexualised bullying or harassment, (ii) bullying or harassment about sexuality, and (iii) bullying or harassment [a]bout gender expression, framing them as dif[f]erent forms of sexual bullying (e.g., Duncan 1999) or gendered harassment (e.g., Meyer 2009), arguing that

they are all underpinned by the performance, reinforcement and enforcement of gender and sexuality norms (Carrera-Fernández et al. 2018; Duncan 1999; Meyer 2009; Renold 2002; Ringrose and Renold 2010).'

Avoiding individualising the problem as bullying discourse tends (Monk 2011), and looking broadly across forms of injustice, we felt our concept might also encompass the different cultural meanings and dynamics in various European countries and regions. The international literature offers a range of terms to identify forms of violence and abuse, with collective categories of violence often indicated, like sexual and gender-based violence (SGBV), sexual orientation and gender identity (SOGI) and sexual orientation and gender-based violence (SOGBV), as well as the debate we had been having since the 1990s about whether misogyny should be viewed as a hate crime (see Gill and Mason-Bish 2013). Campaigns against hate crime can be helpful in the way they illustrate the shared experience of groups of (potential) victims, e.g., London Borough of Tower Hamlets' No Place for Hate campaign which, in an area with a large minority ethnic population (especially Bengali) drew the link between homophobic violence that was generally concentrated outside gay bars and the racial discrimination or violence the majority community might experience. A poster campaign against all hate crime is currently running on the London Underground (Transport For London 2022).

A later project that built on the GAP Work Project and its sister project, USVreact (usvreact/eu) was the SeGReVUni project in which it was decided to retain GRV but extend it to '*sexual and* gender-related violence' (SRGV) which makes explicit the inclusion of sexualised bullying or violence (segrevuni.eu/). SeGReVUni is concerned with the culture among and support for university students and seeks to quantify the problem in order to make it more visible. Another term that research on university campus cultures uses is the term 'sexual and gender-based violence and harassment (SGBVH)' (see Bull et al. 2022, this issue).

## 6. Materials and Methods

The methods employed in the original 'Action Project' are described in the Project's final report (Alldred et al. 2014), and the individual partner Local Action Coordinator training evaluation reports describe the methods used in the evaluations of each country's training courses and related interventions (Biglia et al. 2014; Cullen and Levitan 2014; Inaudi and Turco 2014; McMahon and McArdle 2014). Ethical approval for this study was provided by Brunel University London in the UK, and stated that each of the partners would comply with local expectations of good practice in research ethics and integrity, since not all partners had local ethics boards at the time.

In brief, the 18-month project involved the provision of training to between 180 and 200 youth practitioners in each of the 4 partner countries, with slightly varying structures to the training as suited the varying professional groups. In Italy, health professionals such as nursing staff and hospital doctors were one of the largest groups to attend the training. In Ireland, the training programme was developed for youth and community workers and delivered to those in degree-level training at the university, and was incorporated as an equalities module into the degree programme. In Spain, the programme was more reflexive in tone, making greater use of personal experience to learn from. It was attended mainly by women in teacher and youth-related roles who were, in general, less diverse ethnically than in the UK and Ireland, and it was delivered by feminist activists and professionals.

Local action evaluation reports were completed by the end of project, which means that the reflections by each of the teams on the value of the project were as of the end of the funding period. This does not capture their later views or their post-hoc reflections. There are likely to be many perspectives among 11 partners and 9 associate partners from 6 countries, so this account is necessarily a simplification and is *my* view and my *retrospective* description. Having described the conceptual framework, I will now extract from the different teams' reports their initial conclusions about its relevance and value and then highlight what this might offer the theoretical landscape for studying and eradicating

forms of violence that relate to gender. For more detailed consideration, please see the project's final report (Alldred et al. 2014) and later publications from each team.

## 7. Findings

Having explained the hopes for what the concept of GRV could do, what did it actually do (at this time in these places)? The multiple experiences of using the concept in each of the partner countries are drawn from their accounts written shortly afterwards as noted, although asking *what a concept can do* is a retrospective way of framing the question for me. At the time, I wanted the concept to make cultural norms rather than the victimised individuals the focus. The test cases I had in mind were whether it would capture and problematise femmephobia and machismo in the LGBT+ community ("no fairies, no twinks"), as well as in the cis-gendered and straight community(ies).

Internationally, its success varied and is considered in more detail in Alldred et al. (2014) albeit without the benefit of hindsight. As we discuss in the project's final report, although each of the partner teams used the term GRV at the start of the project, one in practice worked with 'VAWG plus homophobia' and had explicitly shifted to writing about GBV by its final report.

### 7.1. Ireland

The Irish team were Youth and Community Work lecturers and so their action was the development of an enhanced equalities course for Youth and Community Work practitioners in training, and stand-alone training workshops for youth workers already in practice in Ireland. It was delivered primarily to graduate and post-graduate students on the Youth Work and Community Work programmes in the Department of Applied Social Science in Maynooth University, Ireland (aka the National University in Ireland) (McMahon and McArdle 2014). Focus groups held with students and practitioners provided information about their experience of GBV and their training priorities, confirming their desire to interrupt gender oppression in their work. The focused modules were part of a larger course on equality and social justice which allowed a wide range of training methods such as exercises to learn about GBV at the personal, practitioner and trainer level.

The concept allowed them to broaden their approach to well beyond gender equality and VAWG, and although at the start of project team members liked the concept, by the end of the project they were referring to it as gender-based violence (GBV) training, although in practice, still including homophobia. In the Irish context, services such as refuges are sometimes run by the (Catholic) church and hence, support around VAWG is not necessarily in sites free of homophobia. This created tensions for the team who decided that GBV had wider purchase, and shared with the whole Project partners that Catholic dogma was creating value conflict within the project over issues such as women's right to choose to terminate a pregnancy. The project made a sustained impact in Ireland because the training was embedded in the professional education degree, which is still running today.

### 7.2. Italy

In Italy, the training was developed for those health professionals who could be called to work in the domestic violence unit of a large hospital which was identified as an area where greater sensitivity was needed. Health and education professionals attended, with less than 8% of them identifying as men, and most never having had any training on gender equality issues before (Inaudi and Turco 2014). This meant that the 'feminism 101' comment was particularly salient here since it could not be assumed that any equalities training had been attended previously; the course had ambitious aims to cover introductory feminism and to reach queer theory within the course of the programme.

One thing that the concept of gender-related violence did in this context was enable the first collaboration between a Pride (LGBT+) organisation and LGBT+ Centre on the one hand, and, on the other hand, a women's domestic abuse organisation, which in this case, was a network of organisations across the region. This was a detailed collaboration in

which they co-created training combining their two differing perspectives. In addition, the team members' critical awareness of the power of language and of normativity extended beyond gender sensitivity, to the intellectual norms which they identified as embedded in terms used as insults. This meant that they were painfully aware of the abusive language used by senior medics, including at them. We would go so far as to identify this as bullying and verbal harassment, and regrettably, it was present even among personnel drawn in to work on this project. This was identified and problematised at the time, but health settings seem often to be very hierarchical. Indeed, the Italian university seemed so too, and processes here needed the approval of many more professors than for the other partners, but this was then seen to have made a modest feminist contribution by prompting those discussions at 'high levels' in the university. In addition, the project led to the domestic abuse services in the Piedmont region becoming inclusive of those experiencing violence in same-sex relationships. Concern was initially expressed that addressing discrimination against women and discrimination against LGBTQ+ people in the same campaign risked playing down the extent of men's violence against women, shifting the attention of public opinion and policymakers away from the latter, but was assuaged in general and complicated by the recognition of intersecting forms of discrimination like that above. Here, the generally Catholic context shared with Ireland seemed mitigated by the secular and, in fact, medicalised and thus high-status provision of the domestic abuse service, or the project partners' starting point in a health setting. One novelty of the Italian arm of the project was the provision of training at campus level and therefore to staff and students together. It would be interesting to examine the data closely to see if the concept of GRV worked any differently for staff or students as a group. We believe the collaboration had enduring impacts here, although we note the rising right-wing voice in Italy with implications for increasing homophobia and conservative gender politics.

*7.3. Spain*

The Spanish team were based in Catalunya and all their many training resources (videos, slides, an online glossary, etc.) were produced in Catalan as well as Castilian. The Catalan team created a five-session training programme called 'Youth, Gender and Violence: Acting to prevent' that used reflection on personal experience of gendered violence to develop theoretical understandings among participants and to produce activist approaches such as the practice of offering allyship to a violence survivor. Trainees were youth workers, teachers and other youth professionals (e.g., school nurses, social integrators, etc.) working in Barcelona and across the region, and their feedback indicated that they wanted more hours of training and valued an intersectional approach, because they recognised that they were weaker on the politics of race than on sexuality and as a community were less diverse racially (Biglia et al. 2014). This arm of the project impacted on teacher education at multiple universities and had three PhD studies running alongside the funded action (Cagliero 2021; Jiménez Pérez 2022; Olivella Quintana 2016). It emphasised the importance of the width and breadth of the evaluation of the pilot (Biglia et al. 2022) and produced the most prolific set of publications (Biglia and Jímenez 2015; Jímenez et al. 2015, 2016).

The concept of GRV was popular among this team for its fit with their emphasis on sexual diversity and for tackling violence against lesbian and bisexual women. Much time was spent discussing conceptual and definitional issues here and this team were very keen on their term being plural—*violencias de género* (Biglia and San-Martín 2007). This made clear that gender itself is a form of violence because it forces people to fit into a pre-defined, dichotomous construction of identity. Therefore, gender violences are all the forms of violence that are exercised and/or reproduced in gender relations and for social roles. The sex or gender of the subject that exercises or receives the violence/s is therefore irrelevant as even an ungendered body or institution can exercise it. The interconnection between the construction of gender and the heterosexual imperative means that violence against people who are LGBT+ is also understood as an expression of gender violence (Biglia 2015, pp. 28–29).

This enabled a wide understanding of violence that includes, amongst other things, power exercised in relationships, lesbo/homo/transphobia, and violence enacted through institutional, symbolic and community relations. It emphasised the importance of an embodied and intersectional approach and that supporters need to remember that even if a common analysis is possible, the material and emotional consequences of *violencias de género* for people cannot be predicted. What *violencias de género* also did was distinguish this strand of work from other less critical or more mainstream feminist approaches which tend to use the singular form.

### 7.4. United Kingdom

The UK team, based in England, provided training for professionals working with children and young people, who were mostly youth workers and teachers (as Cullen and Levitan 2014 describe), and the educational sessions were themselves animated by the youth work approaches that directly and indirectly[5] informed them, as pedagogies of disruption and discomfort (Cullen and Whelan 2021).

These youth practitioners identified their own behaviour and the development of reflexive practice as key tools in tackling gender-related violence (GRV) and highlighted organisational culture change as necessary, as well as wanting to design educational interventions for use with young people. They identified sexist and homophobic cultures even, in one case, as an element of their own workplace, and this might not have been possible without the overarching framework, since if VAWG and homophobic bullying had been tackled in different training courses, those staff identified as regularly making homophobic comments may well have opted not to attend the latter. The staff members whose views were problematised were senior in the team, so the fact that for this team attendance had been agreed 'en masse' for a local government department was significant, and some members of their team identified the mandatory nature of the training as significant and as likely to be helpful for them in future. At least one youth practitioner was not 'out' to their team because of the homophobic views therein and the supposed justification of these views by the Christian faith.

Overall, attendees' views about the value of the actual term gender-related violence (GRV) were mixed, with some practitioners finding it unnecessarily theoretical and others finding it a helpful link between areas of discrimination and of violence, such as between homophobia and violence against women and girls, as (Cooper-Levitan and Alldred 2022) describes. The problematisation of norms was valued quite widely and it was used to recognise both misogynistic music lyrics and that 'macho' behaviour needed to be problematised whether it was from the young men using the service or the police attending to the young people.

### 8. Discussion

The findings are mixed: in Spain and the UK the concept of GRV seemed helpful, but in Italy and Ireland it was initially expected to be helpful but, in practice, a conceptualisation closer to *gender-based violence plus homophobia* was employed. The tentative conclusion is that where the problematisation of VAWG was well-established *and* LGBT+ rights were too, this framework was productive, but that it was less successful where homophobia and heteronormativity remained in sites that were tackling VAWG. By sites, this could mean cultural, regional sites or specific services that remained heterosexist in culture. One service provider was the Roman Catholic church, but it was not the only one presenting this obstacle.

The teams in Italy, Ireland, Spain and the UK each had different routes to creating contextually relevant training programmes and materials, and worked with different approaches to the recruitment of trainees. Sign-up to the training programmes varied in terms of how the specific professional groups were targeted, and the degree of individual consent or active opting into the training. The UK team managed to provide training for the whole of a London borough's community safety team, but the price of this wide

reach was staff having been compelled to attend the course. The Italian team produced the first training on gender equality for those health professionals, and the partnership they established between a domestic abuse support service that was framed as a women's service and an LGBT+ organisation was novel and created a network that outlived the project.

Internationally, then, its success varied. As we discuss in the project's final report, although most of the partner teams used the term GRV, some in practice tackled 'GBV' or 'VAWG plus homophobia'. The Irish team explain (Alldred et al. 2014, chp. 3) that their preference is for the term 'gender-based violence', and although the other local actions adopt the term gender-related violence, in practice, all four actions problematise (at least) homophobia as well as violence against women and girls.

Furthermore, employing the same term in English might not mean that the actions situated in local contexts and languages have precisely the same definitions and meanings. Indeed, the Spanish team's preference for the plural ('violences') is hard to make visible in English but, in context, this distinguished their intervention from other Spanish approaches that use the singular.

Of course, there are serious limitations to international comparative studies, and caution must be exercised over comparative conclusions. A key issue with this study was its inception from a UK perspective and articulation in English which meant that the terms and concepts used may not have been as relevant for the other countries, and the comfortable translation of the plural 'violencias' into English still eludes me, even though I like its intent (Guizzo et al. 2016). The loose structure of the project was intended to allow the training concept to be rethought in each context, and there will be different answers to the question 'what can a concept do?' for each country's context and also within them. I am grateful to the whole team for engaging constructively with these challenges of translation, equivalence and commonality. There are now new studies exploring the cultural dynamics that are hinted at here (e.g., Flynn 2023; Kondakov 2021; Michelis 2023; Zambelli 2022 etc.) and exploring specific race/gender/sexuality intersectionalities (e.g., Davis 2022; Sundaram et al. 2022).

The tentative conclusion I draw is that where the problematisation of VAWG was well-established *and* LGBT+ rights were too, this framework was successful, but that it was less successful in sites where homophobia and heteronormativity remained, perhaps because these separate 'fights' still felt they needed to battle separately and that there was a cost to allyship with another. However, this could be what makes it a valuable intervention in those sites.

I would be interested to know what this concept can or does do in other geo-political locations. At time of writing, 10 years since the project Call, right-wing mobilization of homophobia and gender conservativism is more visible in many countries in Europe, and in the UK, a Conservative government threatens to review human rights legislation, meaning that future commitment to equalities and human rights cannot be assumed, which would reshape the context for anti-violence interventions such as this.

**Funding:** This research was co-funded by the European Union's Daphne-III Programme (JUST2012/DAP/AG/3176) and led by Dr Pam Alldred at Brunel University London (UK) between 2013-2015.

**Institutional Review Board Statement:** The study was conducted in accordance with the Declaration of Helsinki, and approved (UK action, overall project) by the Brunel University London (UK) (School of Health and Social Care) Research Ethics Committee on 13 January 2013, which also gathered the Maynooth University Ethics Committee approval and statements about local practice from the Italian and Spanish Local Action Coordinators. www.brunel.ac.uk/people/project101293 (accessed on 1 January 2022).

**Informed Consent Statement:** Informed consent was obtained from all participants in Ireland and the UK. In Italy and Spain the study complied with local expectations of good practice at the time.

**Data Availability Statement:** The study did not gain permissions to archive the data.

**Conflicts of Interest:** The author declares no conflict of interest.

## Notes

[1] This research was co-funded by the European Union's Daphne-III Programme (JUST2012/DAP/AG/3176) and led by Dr Pam Alldred at Brunel University London (UK) between 2013–2015.

[2] (https://commission.europa.eu/strategy-and-policy/policies/justice-and-fundamental-rights/gender-equality/gender-based-violence/what-gender-based-violence_en#:~:text=Gender-based%20violence%20is%20violence%20directed%20against%20a%20person,that%20affects%20persons%20of%20a%20particular%20gender%20disproportionately, accessed on 1 January 2022).

[3] The UK team included Dr Fiona Cullen, Malin Elge (nee Stenstrom), Mika Neil Cooper-Levitan, and for a period, Dr Jokin Azpiazu-Carballo and Dr Anna Velasco. Thanks to Malin for reading the Swedish literature. Prof. Miriam E. David and Prof. Ian Rivers were CoI's on the project and because they were London-based supported the UK team particularly.

[4] We are grateful to Professor Erica Burman for introducing us at Manchester Metropolitan University's Discourse Unit.

[5] Since the Local Area Coordinator, Dr Fiona Cullen, was an ex-youth worker and a youth and community work lecturer, and so too was one of the trainers, Dr Michael Whelan.

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
