# Peer review of "Gender-Related Violence: What Can a Concept Do?"

_socsci, doi:10.3390/socsci12090479_

Round 1
Reviewer 1 Report
Thank you for the opportunity to review this manuscript. I enjoyed the invitation to think about the currency and value of the concept of ‘gender-related violence’ and found it useful to hear about how the concept has or has not worked in practice across multiple European countries. I think this manuscript has the potential to make a contribution to the literature; however, there are some substantive issues that I think are important to address. I outline these below.
The concept of GRV
You refer to GRV early on in the manuscript. For example, in the Introduction, you state that this discussion piece offers the concept of GRV, and the first sentence of Section 2 opens with the phrase “Gender-related violence (GRV)”. This continues in Section 3, where you mention that the project aimed to tackle GRV in young people’s lives. However, the reader isn’t given a definition or explanation of GRV until later in the paper. I suggest creating a new section, after Section 2 (Background) and before Section 3 (The Project) that focuses on the concept of GRV. Here, it would be useful to cover: What is ‘GRV’? How do you define it? Have other researchers or practitioners defined it in the same way? Have other researchers or practitioners used similar or different terms? If other terms are commonly used, why do you advocate for this term over another? Why is the term ‘GRV’ important or relevant? What conceptually unites or underpins GRV? Why have you brought different concepts together under the same umbrella?
To some degree, these points are covered in Section 4 (Theoretical Approach), but not all of these points are, and I think Section 4 comes too late in the paper. I would consider removing Section 4, and placing the relevant conceptual information in the new section outlined above, and the material on the imputeus for the project and education package at the beginning of/within Section 3 (The Project), with the paper then going from ‘The Project’ to ‘Materials and Methods’ instead.
Similarly, a lot of the necessary conceptual points are covered in Section 5 (Conceptual Tools), but again, I think this Section comes too late in the paper. I would consider removing Section 5 and integrating the material from it into the new section outlined above, with the paper then going from ‘Materials and Methods’ to ‘Findings’ instead.
Two papers that you might find useful for the new section:
· Frazer & Hutchings (2019), The feminist politics of naming violence: https://journals.sagepub.com/doi/abs/10.1177/1464700119859759?journalCode=ftya
· Turner-Moore, Milnes & Gough (2022), Bullying in five European countries: Evidence for bringing gendered phenomena under the umbrella of ‘sexual bullying’ in research and practice: https://link.springer.com/article/10.1007/s11199-021-01254-1
In relation to the first sentence of Section 2, where you open with the phrase “GRV”, I suggest replacing this with phrases that cover the same meaning for the purposes of this sentence, but without using the term GRV, until you introduce the term in the next section.
Page 7, lines 288-290: It would be useful to explain how the definition of GRV supplied at the bottom of page 6 (lines 282-284) includes intersectional matrices of domination and inequality, particularly around race, ethnicity and class.
Page 7, lines 310-312: Could you explain why the terms SGVB and SOGBV aren’t adequate, and why GRV is more appropriate?
Unique contribution to knowledge and detail about the project and its findings
You often refer the reader to other Project outputs. For me, this raised two issues: firstly, I began to wonder to what extent this paper is making a unique contribution to knowledge? You state that the findings have been reported in more detail elsewhere (Page 8, lines 347-348), so what is the unique aspect of this paper? Perhaps it’s that you are looking back, with the ‘benefit of hindsight’, but if so, I would draw this out and explain it further. It is also worth noting that it is with the benefit of the authors’ hindsight only, for as you explain in Section 4, the PTERs were completed at the end of the project, and it sounds like you didn’t go back to partners for their own thoughts with the ‘benefit of hindsight’?
Secondly, I found that the reference to other Project outputs could mean that this paper was missing useful information for the reader:
· In Section 3 (second paragraph), I was left wondering who the partners were (e.g., were they all NGOs?) and whether there was one partner in each country or not. The information at the end of the later ‘Materials and Methods’ section states there were 11 partners and 9 associate partners from 6 countries, so I’m assuming there was more than one partner per country? In Section 3 (second paragraph), I was also left wondering who the trainees were (e.g., were they all youth practitioners?) and how many trainees there were in each country. Later, I read some of this information in the Findings section, but the Method section started to raise these questions earlier for me.
· In Section 4 (Materials and Methods), I would like to know more about the PTERS. They were completed by the partner at the end of the project but what material did the partners draw on to inform their reports? This seems relevant since you are purportedly drawing on the findings from these reports later in the paper.
· In Section 6 (Findings), I found it informative to hear about how the term ‘GRV’ was or was not used in each country and the reasons for and impacts of this. However, this did feel a little brief.
· In Section 7 (Discussion) line 482, you state “as we discuss in the project’s final report”, but given the salience of this material, I think the points need to be covered in the paper itself.
Brief and less developed Discussion section
The Discussion section seemed a little brief and I think this could be developed more. This could include unpacking the existing points in more detail, and introducing new points. Some of the remaining questions I had when reading this Section were:
· What were the problems (if any) with the partners using ‘GBV’ or ‘VAWG plus homophobia’ or ‘gender violences’ instead? Do you think this was a ‘good/valuable’ equivalent to ‘GRV’ or even a ‘better’ alternative in these contexts?
· What are your thoughts and reflections on the potential for use of the term ‘GRV’ globally (i.e., beyond Europe)?
· How do your findings relate back to the more conceptual arguments you make earlier in the paper about using the term ‘GRV’? I would end the Discussion by returning to the material that you will cover in the new section at the beginning of the paper, where you introduce the concept of ‘GRV’. Based on the findings from your project, would you still advocate for the term ‘GRV’? Does the term ‘GRV’ make the conceptual/knowledge contribution that you thought it would, as set out earlier in the paper? Currently, in the Discussion section, you cover the practice contribution of the term, but it would be informative to then return to more conceptual issues, to round off the paper.
Minor issues
I recommend proof-reading throughout; I spotted a few minor errors throughout. For example, lines 29, 52, 88, 98 ‘Seal PhD’, 99 (no year for Stonewall), 249, 303, 363, 395, 477.
There are two Section 4s!
Section 5: Check for missing supporting references (e.g., in opening paragraphs). Also, Section 6.3, what’s the reference for the quote?
Check that page numbers are consistently given for quotations.
Page 7, line 328: Should there be ‘it’ after ‘has proven’?
Reviewer 2 Report
It is an interesting article that describes the experience of training on gender violence in practitioners who work with young people. Although their objectives say that they focus on conceptual development or how it has emerged, some gaps make it difficult to assess the work.
1. The problem is not entirely clear. Is it a knowledge gap in treating gender violence in young people? The literature review needs to demonstrate that there is a knowledge gap and describe other significant advances.
2. The program description should be more detailed, especially with country adaptations. The method section is very superficial. It would be necessary to describe how many groups there were, how many people, the number of hours of training, and modalities, among others. The qualitative method of comparative analysis is also absent, which makes it difficult to assess the rigor of the paper.
3. In the discussion, do not compare the superiority of their proposal compared to other training proposals.
Reviewer 3 Report
This paper presents findings from a pan-European project as the basis for testing a revised approach to gender-based violence, re-introducing the concept of gender-related violence (GRV). The ideas contained within the paper are very relevant and of interest to those working in the wider VAWG / LGBTQIA+ arenas. However, there are some areas in the paper where the ideas could be expressed better / presented differently to ensure clarity and understanding.
The first paragraph juxtaposes the barriers faced by children and young people seeking to explore and/or express their sexual or gender identity with theoretical insights on violence. Is the intention here to see these barriers as a form of violence? Certainly, this is a perspective held by some, but to those unfamiliar with perceiving violence in this way, it may be considered a strong claim and inferring malicious intent on the part of those interacting with these children and young people. More clarification would be good to indicate whether the paper is considering such barriers as synonymous with verbal, physical or other forms of violent conduct, or as being on a trajectory with them. [Added to note that you appear to address this later on in lines 219-223, so perhaps this should come earlier in the paper.]
The reference to young people's experience of online violence and abuse (L78) is relevant, but would be strengthened with some reference to the multitude of studies addressing this issue. In particular, Dragiewicz's work would be good to cite here.
There are some taken-for-granted concepts included (but not explained) in the paper. Perhaps the audience is intended to be those working / researching in this area, but that might limit the scope of the ideas included within it. Concepts like 'gender normativity' and the Gender Order may be less familiar to some, for example.
The paper is written in a somewhat reflective manner, which is fine, but at times the subjective presentation of information seems inappropriate and as though the reader is being 'guided' around what to think. For example, in L204 the author talks about prejudice within the LGBT+ community as being 'sad' to recognise, implying that perhaps this insight was new to the author, or that they thought LGBT+ communities had moved beyond this intra-prejudicial bias (the existence of which has been noted in several sexualities studies). To add greater credence to the overall message contained within this paper, I'd recommend reconsidering how such points are presented.
L256, where you indicate that 'some' made the links between the UK women's and gay liberation movements, citing some sources would be useful here as the more well-known ones tend to be US based.
The omission of full methodological information is accounted for, but in doing so the findings are harder to assess / compare as no information is given about size of cohort, the length of time data was collected etc. Presumably, these reflective insights are to be read separately, but the discussion then seeks to find correlations and contradictions between them in order to assess the salience of employing GRV as a concept. I'm not sure what to suggest here, but perhaps when the links to the related project papers are included this will provide the necessary insight to make sense of how comparable these projects were.
Some editorial pointers:
Generally:
There appears to be some inconsistent capitalisation throughout the paper, unless there's a reason for this which is not evident.
There's some outstanding references which appear to need including (not the anonymised ones).
There's an inconsistent use of LGBT+, LGBTQ, LGBTQ+ etc.
Specifically:
Typo in the first keyword
Some of the language needs revision in places. For example, lines 47-48 replace 'was wanted' with 'was sought' or 'was desired'.
The sentence on lines 82-87 is too long.
The sentence on line 94 reads as unfinished and the sentence after this needs clarification to indicate who did this acknowledging.
Similarly, line 101, recognised by whom?
L142: 'norm critical pedagogies' reads a little strangely. Do you mean orthodox approaches? Also, I'm not sure if you mean these were a positive or negative influence on the project design.
L166: typo 'on for generic'
L190-1: a word missing here, 'any' or 'a' chance
L197: some reference to the website where these are held would be useful here, yes.
L303: is the sentence on identity politics correct / complete?
L330: unclear where this quotation starts
L337: This sentence reads a little strangely so would benefit from being revised for clarity
L360: it's 'of Ireland' not in Ireland
L363: is 'formation' the correct word (not foundation)? If so, a formation programme needs explaining.
L395: personnel, not personal
L407: Remove the 'In fact' from the start of the sentence
L499: loose, not lose
Round 2
Reviewer 2 Report
Well done.